# Reparative Efficacy of Liposome-Encapsulated Oleanolic Acid against Liver Inflammation Induced by Fine Ambient Particulate Matter and Alcohol in Mice

**DOI:** 10.3390/pharmaceutics14051108

**Published:** 2022-05-23

**Authors:** Ching-Ting Wei, Yu-Wen Wang, Yu-Chiuan Wu, Li-Wei Lin, Chia-Chi Chen, Chun-Yin Chen, Shyh-Ming Kuo

**Affiliations:** 1Department of Electrical Engineering, I-Shou University, Kaohsiung 82445, Taiwan; triplet0826@gmail.com; 2Department of Biomedical Engineering, I-Shou University, Kaohsiung 82445, Taiwan; yuwen870928@gmail.com (Y.-W.W.); memtal0825@gmail.com (C.-Y.C.); 3Department of Surgery, E-Da Hospital, I-Shou University, Kaohsiung 82445, Taiwan; 4Kaohsiung Armed Forces General Hospital, Kaohsiung 81342, Taiwan; ranger.wu1113@gmail.com; 5Ministry of National Defense-Medical Affairs Bureau, Taipei City 10462, Taiwan; 6The School of Chinese Medicine for Post Baccalaureate, I-Shou University, Kaohsiung 82445, Taiwan; lwlin@isu.edu.tw; 7Department of Pathology, E-Da Hospital, I-Shou University, Kaohsiung 82445, Taiwan; sasabelievemydream@gmail.com

**Keywords:** oleanolic acid, liposome, PM_2.5_, alcohol, liver inflammation

## Abstract

Airborne fine particulate matter (PM_2.5_) is a severe problem and is associated with health issues including liver diseases. Workers performing manual labor tend to be alcohol consumers during work, where they are also exposed to PM_2.5_. Long-term PM_2.5_ exposure can increase oxidative stress, leading to inflammation. Whether long-term exposure to air pollution and alcohol synergistically increases liver fibrosis risk warrants investigation. Oleanolic acid (OA)—a triterpenoid—has antioxidant and anti-inflammatory activities, but its low water solubility and cytotoxicity impair its potential applications. In this study, we fabricated liposomal OA nanoparticles (Lipo-OAs); then, we evaluated the anti-inflammatory effect on exposed cells and the ameliorative effect of Lipo-OAs on PM_2.5_ and alcohol-induced liver fibrosis in mice. The half maximal inhibitory concentration of PM_2.5_ for hepatic stellate cells was 900 μg/mL; at a concentration of ≥600 μg/mL, PM_2.5_ significantly increased interleukin-6 and tumor necrosis factor-α production. OA encapsulation in Lipo-OAs, 353 ± 140 nm in diameter with 79% encapsulation efficiency, significantly reduced OA cytotoxicity. Lipo-OAs treatment significantly reduced alanine aminotransferase, aspartate aminotransferase, and γ-glutamyltransferase levels; histologically, it alleviated steatosis and improved Ishak’s modified HAI score. In conclusion, Lipo-OAs have potential anti-inflammatory and reparative effects for PM_2.5_ and alcohol-induced liver injury treatment.

## 1. Introduction

Liver diseases are major health issues globally. Chronic liver injury can result in fibrosis and cirrhosis or can even progress to hepatocellular carcinoma. In the liver, fibrosis is a self-protecting mechanism in the wound healing process. In response to liver injury, hepatic stellate cells become activated and produce fibrotic scars at the injury site [1]. Factors such as viral infection and chronic alcohol consumption can damage hepatic stellate cells; consequently, a self-repair process is initiated in hepatic stellate cells, causing scar tissue to accumulate, and ultimately leading to liver fibrosis.

Currently, air pollution is a severe problem. One of its consequences is smog, the main component of which is fine ambient particulate matter (aerodynamic diameter < 2.5 μm; PM_2.5_). Long-term PM_2.5_ exposure can produce excessive oxidative stress in cells and cause inflammation. Exposure to PM_2.5_ is a risk factor for pulmonary and cardiovascular diseases, as well as metabolic syndrome [2,3]. Traffic-related airborne PM_2.5_ is a complex mixture of particles and gases from gasoline and diesel engines, as well as dust from road surfaces, tires, and brake wear [4,5]. With diminishing sizes, airborne PM_2.5_ has an incremental capacity to penetrate the distal airway units and potentially enter the systemic circulation. The cytotoxic effect of PM_2.5_ is more associated with PM_2.5_ as a complex than with single or a few PM_2.5_ components [6]. The particle sizes, charges, and combined effects of individual PM_2.5_ components are crucial contributors to the adverse health impact of PM_2.5_ exposure. PM_2.5_ exposure may trigger various maladaptive signaling pathways in the lungs, blood vessels, liver, and adipose tissue that are associated with endoplasmic reticulum stress, oxidative stress, and inflammatory responses [7,8,9]. Moreover, in animals, long-term PM_2.5_ inhalation has been found to cause a nonalcoholic steatohepatitis (NASH)-like phenotype and hepatic glycogen storage depletion [10]. Furthermore, PM_2.5_ exposure promotes NASH-associated activities and impairs hepatic glucose metabolism. A study reported that exposure to PM_2.5_ for 10 weeks led to hepatic lipid–glucose homeostasis disruption, lobular and portal inflammation, and mild hepatic steatosis in the mouse liver [11].

Excessive alcohol consumption can cause liver-related diseases. A fatty liver lesion—the first hepatic response to long-term alcohol consumption—might become alleviated after alcohol withdrawal [12]. However, in some severe liver injuries, such as liver inflammation, liver fibrosis, and steatohepatitis may persist and impair liver function, even after alcohol withdrawal. When at work, individuals employed in manual labor jobs tend to not only consume alcohol but also be exposed to severe air pollution. Hence, whether excessive alcohol consumption and long-term air pollution exposure synergistically affect the liver to ultimately cause liver fibrosis warrants investigation. Currently, of the many liver-protective medications and foods, herbal medicines (i.e., those including plant extracts) are gradually replacing synthetic pharmaceuticals, and they are considered complementary, alternative medications.

Oleanolic acid (OA) is a naturally occurring pentacyclic triterpenoid and is widely distributed in plants. It can act as an aglycone precursor for triterpenoid saponins and link to one or more sugar chains [13]. OA and its derivatives exhibit several biological activities, including hepatoprotective, anti-inflammatory, and antioxidant effects. OA was recently used as a base molecule for its modification into potential antitumor and antifibrosis agents [14,15]. With the recent elucidation of its biosynthesis and the imminent commercialization of the first OA-derived drug, the compound has promising applications. Being a natural product, OA has been widely used for hepatopathy treatment in China; however, its clinical applicability remains elusive due to its low water solubility. Considering the marked bioactivity of triterpenoid saponins, OA synthesis and biological evaluation have attracted considerable attention.

Liposomes, composed of one or more lipid bilayers, are structured similarly to cell membranes; hence, they are biocompatible and biodegradable. Liposomes have thus been widely used as carriers for drugs; their low bioavailability and water solubility improve drugs’ pharmacodynamics and therapeutic efficacy. In our previous study, we prepared 1,2-distearoyl-sn-glycero-3-phosphocholine (DSPC)-based liposomes to encapsulate astaxanthin—which has low bioavailability and water solubility—to treat alcohol-induced liver fibrosis. These astaxanthin liposomes had protective effects and therapeutic potential, demonstrating that liposome encapsulation increased astaxanthin bioavailability and potency in vivo [16,17,18].

In the present study, we fabricated liposomal OA nanoparticles (Lipo-OAs) and evaluated OA and Lipo-OAs’ toxicity in hepatic stellate cells through the MTT cell viability assay and flow cytometry apoptosis assay. Their anti-inflammatory and antioxidant effects on the hepatic stellate cells exposed to PM_2.5_ were also evaluated. Finally, the efficacy of the protective and reparative effects of OA and Lipo-OAs in mice with PM_2.5_ and alcohol-induced liver fibrosis were examined and verified through histopathology, Western blotting, and serum alanine aminotransferase (ALT), aspartate aminotransferase (AST), total bilirubin, and γ-glutamyl transferase (GGT) analyses.

## 2. Materials and Methods

### 2.1. Materials

OA, cholesterol, DSPC (molecular weight, 790.15 Da), chloroform, and methanol were obtained from Sigma (St. Louis, MO, USA). PM_2.5_ was purchased from the National Institute of Standards and Technology (Urban Dust-1694b, Gaithersburg, MD, USA). All chemicals used in the experiment were of reagent grade. The animal research was approved by the Institutional Animal Care and Use Committee of I-Shou University, Taiwan (approval NO: IACUC-ISU-108034, approval date: 2 December 2020).

### 2.2. Lipo-OAs Production and Characterization

Lipo-OAs were prepared using the evaporation sonication method with some modifications [4]. In brief, liposomes were first prepared by dissolving the phospholipid DSPC (8.7 mg), cholesterol (2.9 mg), and OA (6.85 mg) in methanol–chloroform (1:3, *v*/*v*). This mixture was then homogenized for 120 s (UP 200S, Berlin, Germany) and dried in a rotary evaporator (N-1300, Tokyo, Japan) into a thin film. This film was rehydrated with deionized water (1 mL) and sonicated for 20 min (Figure 1a). The Lipo-OAs size was calculated by randomly sampling approximately 150 liposomes under a transmission electron microscope. The encapsulation efficiency of OA in the liposomes and the in vitro release of OA from the liposomes were analyzed according to our previously reported method [19,20,21].

### 2.3. In Vitro Cell Viability Tests of OA and Lipo-OAs

Rat LX-2 cells (cell line, purchased from Elabscience, Minneapolis, MN, USA) were suspended at a density of 7 × 10^3^ cells/mL in 96-well plates containing Dulbecco’s modified Eagle’s medium supplemented with 100 U/mL penicillin, 100 µg/mL streptomycin, and 10% fetal bovine serum (Gibco). After the cells were incubated at 37 °C and 5% CO_2_ for 24 h, the medium was replaced with fresh medium containing various concentrations of OA, Lipo-OAs, and PM_2.5_. LX-2 cell viability was examined using the MTT assay. At 24 h after treatment with OA and Lipo-OAs, 20 μL of MTT solution was added to the cells, followed by incubation for an additional 3 h. The formazan precipitate was dissolved in 200 μL of dimethyl sulfoxide through vertexing; next, the absorbance of this solution was measured at 570 nm on a multiplate reader (Thermo Scientific, Waltham, MA, USA).

Cell viability was also examined using the flow cytometry apoptosis assay (BD Biosciences, San Diego, CA, USA). In brief, LX-2 cells were seeded at a density of 6 × 10^4^ cells/well in 24-well plates and then incubated at 37 °C and 5% CO_2_ for 24 h. The medium was then replaced with fresh medium, and OA or Lipo-OAs were added at various concentrations to each well, followed by incubation for 24 h. Next, a staining kit containing FITC-conjugated annexin V/PI was used, according to the manufacturer’s instructions. In brief, the cells were detached from the culture flask through trypsinization and washed with phosphate-buffered saline. Cell pellets were suspended in 1× binding buffer and stained with 5 μL of annexin V-FITC and 10 μL of PI for 15 min. The cells were then subjected to fluorescence-activated cell sorting analysis on a flow cytometer (BD Accuri C6, Franklin, NJ, USA).

### 2.4. In Vitro Cell Viability and Inflammatory Response of LX-2 Cells Exposed to PM_2.5_

The in vitro cell viability test was conducted as described in the previous section. The inflammatory response of LX-2 cells to PM_2.5_ exposure was examined as follows: LX-2 cells were seeded at a density of 2 × 10^5^ cells/well in 6-well plates and incubated at 37 °C and 5% CO_2_ for 24 h. Subsequently, the cells were treated with various PM_2.5_ concentrations (200–1000 μg/mL). After cell incubation for 24 h, the medium was collected and centrifuged at 2000× *g* for 15 min. The supernatant was then assayed for interleukin (IL)-6 and tumor necrosis factor (TNF)-α using Elabscience human ELISA kits, and absorbance was measured at 450 nm, following the manufacturer’s protocol.

### 2.5. Antioxidant and Albumin Secretion by PM_2.5_-Induced Inflammatory LX-2 Cells Treated with OA and Lipo-OAs

The antioxidant capability of OA and Lipo-OAs on LX-2 cells with PM_2.5_-induced inflammation was determined using the free radical-scavenging activity assay and the H_2_O_2_ scavenging test. The LX-2 cells (6 × 10^4^ cells/mL) were seeded in a 24-well plate. After 24 h of incubation, the cells were treated with 600 μg/mL PM_2.5_ (inflammatory concentration for LX-2 cells; results are shown in Figure 2) and were incubated for 24 h to induce an inflammation response in LX-2 cells. After incubation for 24 h, the cells were treated with OA and Lipo-OAs at various concentrations. After cell incubation for another 24 h, H_2_O_2_ concentrations were determined by measuring absorbance at 570 nm on the multiplate reader.

An albumin secretion assay was performed next, as described elsewhere [22]. The medium was collected from the cultured samples, and the secreted albumin levels were quantified using ELISA kits (BioVision, Milpitas, CA, USA) following the manufacturer’s instructions.

### 2.6. Animal Experiments

#### 2.6.1. Establishment of PM_2.5_ and Alcohol-Induced Liver Inflammation Model in Mice

In total, 51 C57BL/6J male mice (age = 6 weeks) were used to establish a PM_2.5_ and alcohol-induced liver inflammation model. The mice were randomized into two groups. Both groups 1 and 2 included 24 mice each; the mice were placed into an electric fan-equipped chamber (created in-house) containing a considerable amount of PM_2.5_ for 4 h for inflammation induction. The group 2 mice were then fed with 100 μL of 30% alcohol. The induction was performed per day for 3 weeks for both of the groups. After inflammation induction, the mice were treated with three doses per week of 0.15 mM OA, 0.15 mM Lipo-OAs, or pure liposomes intraperitoneally (i.p.) for 4 or 6 weeks. Even during the OA, Lipo-OAs, or pure liposome treatment, the mice remained under inflammation induction treatment. In addition, a control group containing three healthy mice was included. In summary, the in vivo animal experiments were performed for up to 9 weeks (Figure 3): In the first 3 weeks, we focused on developing our PM_2.5_ and alcohol-induced liver inflammation model; this was followed by OA, Lipo-OAs, or pure liposome treatment for 4 and 6 weeks.

After all of the treatments, the mice were sacrificed, and their livers and blood samples were collected to perform histopathological, Western blot, and serum biochemical analyses. In all of the experiments, the mice were anesthetized with tiletamine + zolazepam (40 mg/kg) and xylazine (10 mg/kg) i.p. During the in vivo animal study period, clinical signs of pain, salivation, and abnormal behavior were monitored.

#### 2.6.2. Histopathological Analysis

After 4 and 6 weeks of treatment with OA, Lipo-OAs, or pure liposomes, the mice were sacrificed, and the harvested livers were fixed in 10% neutral-buffered formalin, dehydrated in graded alcohol solutions, cleared in xylene, embedded in paraffin blocks, and cut into 5-µm thick sections. This was followed by hematoxylin and eosin (HE) staining and Masson’s trichrome staining for histopathological examination.

#### 2.6.3. Ishak’s Modified HAI Score

Ishak’s modified HAI score [23,24] was used to evaluate the liver necrosis and inflammation severity. The score was determined by surgeons and pathologists, based on four components:(1)Periportal or periseptal interface hepatitis score: 0 = absent; 1 = mild (focal, few portal areas); 2 = mild or moderate (focal, most portal areas); 3 = moderate (continuous around <50% of tracts or septa); and 4 = severe (continuous around >50% of tracts or septa);(2)Confluent necrosis score: 0 = absent; 1 = focal confluent necrosis; 2 = zone 3 necrosis in some areas; 3 = zone 3 necrosis in most areas; 4 = zone 3 necrosis + occasional portal–central (P–C) bridging; 5 = zone 3 necrosis + multiple P–C bridging; and 6 = pan-acinar or multiacinar necrosis;(3)Focal (or spotty) lytic necrosis or apoptosis and focal inflammation score: 0 = absent; 1 = ≤1 foci per 10× objective field; 2 = 2–4 foci per 10× objective field; 3 = 5–10 foci per 10× objective field; and 4 = >10 foci per 10× objective field;(4)Portal inflammation score: 0 = none; 1 = mild (some or all portal areas); 2 = moderate (some or all portal areas); 3 = moderate or marked (all portal areas); and 4 = marked (all portal areas).

#### 2.6.4. Western Blot Analysis

To examine the inflammation alleviating and protective effects of OA and Lipo-OAs treatments, we assayed the mouse livers for smooth muscle α-actin (αSMA), IL-6, and TNF-α. In brief, polyvinylidene difluoride membranes were prewetted and then immersed in transfer buffer for 30 min. The proteins from our samples were transferred and blotted onto these membranes. The membranes were then incubated in Tris-buffered saline with 5% skim milk powder and 0.05% Tween 20 for 1 h at room temperature to block nonspecific protein binding. The membranes were then incubated with primary antimouse antibodies for αSMA, IL-6, TNF-α, and GAPDH overnight at 4 °C. The ratio of the αSMA, IL-6, TNF-α, and GAPDH protein levels in the experimental groups to those in the control group was measured through semiquantitative intensity analysis (normalized by the respective GAPDH and background) by using ImageJ (Version 1.50; National Institutes of Health, Bethesda, MD, USA).

#### 2.6.5. Blood Biochemical Assays

Blood biochemical parameters, including ALT, AST, total bilirubin, and GGT, were assayed to evaluate liver function repair after treatment with OA, Lipo-OAs, or pure liposomes in the mice with PM_2.5_ and alcohol-induced liver damage. Serum was isolated from the whole blood sample and subjected to ALT, AST, and GGT assays, according to the manufacturer’s instructions (AAT Bioquest, Sunnyvale, CA, USA).

### 2.7. Statistical Analysis

The data are presented as means ± standard deviations. The experimental data were pooled from three independent experiments. Significant differences (indicated by *p* < 0.05) between the control and experimental groups were determined with SPSS by using one-way analysis of variance.

## 3. Results

### 3.1. Lipo-OAs Characterization

Our Lipo-OAs had a diameter of 353 ± 140 nm, favorable sphericity (Figure 1b), and 79% OA encapsulation efficiency. Moreover, approximately 54% of OA was released from the Lipo-OAs after incubation at 37 °C for 24 days on a 40-rpm shaker (Figure 1c).

### 3.2. Effects of OA and Lipo-OAs on Cell Viability

The effects of OA and Lipo-OAs on LX-2 cell viability were assessed using the MTT assay and apoptosis assay, as shown in Figure 2. The half maximal inhibitory concentration (IC_50_) of OA in LX-2 cells was approximately 0.15 mM; notably, the IC_50_ of Lipo-OAs was considerably higher at approximately 0.25, indicating that the OA cytotoxicity decreased after encapsulation and yielded more flexibility in using the OA chemicals.

The effects of OA and Lipo-OAs on LX-2 cells were monitored under a light microscope. The cells initially adhered and spread well onto the plates when the OA concentration was <0.1 mM; however, they gradually shrank into a round shape and detached from the plates, indicating cell death. Their pseudopodia shrank gradually under treatment with 0.3 mM Lipo-OAs, indicating that the Lipo-OAs have low cytotoxicity; these results were consistent with those of the MTT assay (Figure 2b).

Next, to examine the toxicity reduction due to OA encapsulation, the apoptosis of LX-2 cells treated with OA and Lipo-OAs at various concentrations was analyzed. As shown in Figure 2c, after encapsulation, the OA cytotoxicity decreased significantly even at a concentration of >0.15 mM. LX-2 cell apoptosis after 0.3 mM Lipo-OAs treatment decreased significantly to one-third of that after 0.3 mM OA treatment. All the cell viability assay results indicated that encapsulation effectively reduced the OA cytotoxicity; moreover, the slow release of OA from Lipo-OAs provided a sustainable OA stimulus. The effects of various concentrations of OA and Lipo-OAs on the cell viability can be seen in the Appendix A.

### 3.3. Effects of PM_2.5_ on Cell Viability and Inflammatory Response

The in vitro effects of PM_2.5_ on LX-2 cell viability and inflammatory responses are shown in Figure 4. The LX-2 cell viability significantly decreased after treatment with >600 μg/mL PM_2.5_, with the IC_50_ of PM_2.5_ being approximately 780 μg/mL for LX-2 cells (Figure 4a). On treatment with >600 μg/mL PM_2.5_, the cells lost their normal spindle shape and gradually shrank when they came in contact with the aggregated PM_2.5_ particles (Figure 4b). Moreover, after treatment with 600 μg/mL PM_2.5_, the cells produced significant amounts of IL-6. Similarly, exposure to >1000 μg/mL PM_2.5_ led to significantly increased TNF-α production in LX-2 cells (Figure 4c,d). These results indicate that PM_2.5_ reduces the LX-2 cell viability, and that, at concentrations of 600–1000 μg/mL, it induces an inflammatory response.

Figure 4e illustrates that exposure to PM_2.5_ led to increased H_2_O_2_ production in LX-2 cells, indicating that exposure to excessive PM_2.5_ concentrations induces free radical synthesis. Nonetheless, we found that 0.05–0.3 mM OA or Lipo-OAs had a significant H_2_O_2_-scavenging capacity in LX-2 cells. The anti-inflammatory effect of OA was shown in Figure 4f,g, the IL-6 and TNF-α levels were significantly suppressed after being treated with OA. However, the Lipo-OAs exhibited lower anti-inflammatory effects, compared with pure OA. These results might be attributed to the few OAs that were released from the Lipo-OAs and which then exerted minor effects on the PM_2.5_-induced inflammatory cells.

Under physiological conditions, albumin—typically synthesized by liver hepatocytes—has significant antioxidant potential. In other words, albumin is involved in the scavenging of free radicals, which are implicated in the pathogenesis of several inflammatory diseases [25]. Thus, it is often considered to be a crucial liver function indicator. As shown in Figure 4h, 600 μg/mL PM_2.5_ exposure led to a significant decrease in albumin synthesis in LX-2 cells. In the LX-2 cells with PM_2.5_-induced inflammation, OA or Lipo-OAs treatment restored albumin secretion to a range similar to that in the control group. Moreover, the higher the OA and Lipo-OAs’ concentration, the more significant was the increase in albumin secreted—indicating that OA can stimulate albumin synthesis in LX-2 cells. The effects of various concentrations of PM_2.5_ on the cells’ inflammatory responses can be seen in the Appendix A.

Considering the cell viability and inflammatory response-related results in LX-2 cells, we used 600 μg/mL PM_2.5_ for the subsequent in vivo animal experiments.

### 3.4. Histopathological Analysis of Mouse Liver

Figure 3 illustrates the timeline and procedures used for the in vivo animal studies. The period for PM_2.5_ induction (i.e., PM_2.5_-induced liver inflammation) was 3 weeks, which was followed by continued exposure to PM_2.5_ with or without OA or Lipo-OAs treatment for the next 4 or 6 weeks. Similar exposure procedures were conducted in the PM_2.5_ and alcohol-induced liver inflammation group. OA and Lipo-OAs concentrations were set at 0.15 mM, according to the IC_50_ of OA in LX-2 cells.

The protective and reparative effects of OA and Lipo-OAs on PM_2.5_ and alcohol-induced liver inflammation and injury in vivo were evaluated through histopathological analysis. Figure 5A presents the HE and Masson’s trichrome staining results of normal mouse liver. We did not replicate the other studies using thioacetamide or carbon tetrachloride to induce liver injury [26,27], and used PM_2.5_ and alcohol induction to simulate the workers’ environment instead. Figure 5b shows the HE and Masson’s trichrome staining major results for the mice who received PM_2.5_ exposure for 3 weeks and treatment with Lipo-OAs for the next 4 weeks. The livers with PM_2.5_-induced inflammation exhibited focal steatosis and diffuse pericellular fibrosis, as observed through Masson’s trichrome staining. Treatment with OA and Lipo-OAs for 4 weeks led to decreases in pericellular fibrosis, compared with no treatment and pure liposome treatment, indicating the repair effect of OA; moreover, the mice treated with pure liposomes had more steatosis and a higher Ishak’s modified HAI score (Table 1). After 6 weeks of Lipo-OAs treatment, the mice exhibited less steatosis, fewer Mallory bodies and ballooning cells, and lower Ishak’s modified HAI scores than did the other mice (Figure 5c and Table 2). The HE and Masson’s trichrome staining results of PM_2.5_-induced inflammation liver treated with OA, Lipo-OAs, and pure liposomes can be seen in the Appendix A. The results indicated that the livers with PM_2.5_-induced inflammation exhibited more pericellular fibrosis after the 7-week period (3 induction weeks + 4 treatment weeks), and the histopathological findings indicated chronic inflammation with fibrosis (for nonalcoholic steatohepatitis), beginning in the pericellular area. Taken together, 6 weeks of Lipo-OAs treatment downgraded the NASH (non-alcoholic steatohepatitis) to NAFLD (non-alcoholic fatty liver disease) [28,29].

Compared with the liver with PM_2.5_-induced inflammation, the liver with PM_2.5_ and alcohol-induced inflammation showed more diffuse steatosis, more Mallory bodies and ballooning cells, and worse Ishak’s modified HAI score (Figure 6). The increase in the number of Mallory bodies and ballooning cells was mainly associated with alcohol-induced liver disease, which worsened to a more severe liver injury under PM_2.5_ and alcohol-induced inflammation [30]. After treatment with OA and Lipo-OAs for 4 weeks, steatosis and the number of Mallory bodies and ballooning cells decreased in the inflamed liver, indicating the reparative effect of OA. Lipo-OAs, nonetheless, further improved the reparative effect, leading to an even lower Ishak’s modified HAI score (Figure 6a, Table 3). As mentioned earlier in the text, pericellular fibrosis due to alcohol consumption was visible focally after Lipo-OAs treatment. As the treatment period extended to 6 weeks, the Lipo-OAs-treated mice exhibited further decreases in the number of Mallory bodies and ballooning cells and Ishak’s modified HAI scores. Diffuse steatosis appeared in the inflamed livers, which was potentially attributable to PM_2.5_ and alcohol induction over a long period (Figure 6b, Table 4). The HE and Masson’s trichrome staining results of PM_2.5_ and alcohol-induced inflammation liver treated with OA, Lipo-OAs, and pure liposomes can be seen in the Appendix A.

In summary, Lipo-OAs treatment led to improved reparative effects—with fewer Mallory bodies and ballooning cells, less pericellular fibrosis, and lower Ishak’s modified HAI scores in the inflamed liver—compared with the other groups. The trend of lower steatosis was also observed after 4 weeks of Lipo-OAs treatment (Table 3).

### 3.5. Western Blot Analysis

The presence of the inflammatory factors αSMA, IL-6, and TNF-α in the mouse liver was visualized through Western blotting. As shown in Figure 7a, the untreated groups (continually exposed to PM_2.5_ for 7 weeks) had the highest αSMA, IL-6, and TNF-α levels. Moreover, 0.15 mM Lipo-OAs exerted higher inhibitory effects on these inflammatory factors than did 0.15 mM OA and pure liposomes. However, 0.15 mM OA or 0.15 mM Lipo-OAs led to slight or no obvious suppression of αSMA, IL-6, and TNF-α production in the livers with PM_2.5_ and alcohol-induced inflammation, even after 4 and 6 treatment weeks (Figure 7b).

### 3.6. Biochemical Analysis of Serum ALT, AST, and GGT Levels

Serum ALT, AST, and GTT levels were examined next, to validate the liver function of the mice treated with OA or Lipo-OAs (Figure 8). The results revealed much higher ALT, AST, and GGT levels in the livers with PM_2.5_ and alcohol-induced or PM_2.5_-induced inflammation than in the normal livers—indicating that exposure to PM_2.5_, especially along with alcohol exposure, impairs liver function and increases the ALT, AST, and GGT levels. The ALT, AST, and GGT levels significantly decreased after 4 and 6 weeks of treatment with OA or Lipo-OAs. In particular, Lipo-OAs reduced the ALT, AST, and GGT levels more significantly than did OA and pure liposomes, possibly because of the slow release of OA from Lipo-OAs, which led to sustained anti-inflammatory effects.

In summary, the observations for the ALT, AST, and GGT levels after Lipo-OA treatment were generally consistent with the findings of the histopathological analysis.

## 4. Discussion

Many reports have indicated that long-term airborne PM_2.5_ exposure can induce inflammatory responses on the skin and in the respiratory system, even leading to hair loss due to the excessive oxidative stress [31,32]. However, few studies have focused on the effects of PM_2.5_ on liver-related diseases. Furthermore, workers who perform manual labor often consume alcoholic drinks at work; thus, whether long-term exposure to alcohol and air pollution synergistically increases liver inflammation warrants investigation. In the current study, we used the liposomal encapsulation technique to reduce OA toxicity and investigated the reparative efficacy of liposome-encapsulated OA on PM_2.5_ and alcohol-induced liver injury in mice. The results indicated that after liposomal encapsulation, OA toxicity decreased significantly, and its antioxidant and liver function preservative characteristics were retained (Figure 3e,f). The IC_50_ concentration of PM_2.5_ in LX-2 cells was approximately 900 µg/mL; moreover, at 600 µg/mL, PM_2.5_ increased the IL-6 and TNF-α production in these cells.

Under physiological conditions, albumin has considerable antioxidant potential. Because it is involved in free radical scavenging, it is often used as a crucial liver function indicator [25]. Here, we found that in LX-2 cells with PM_2.5_-induced inflammation, OA and Lipo-OAs restored albumin secretion to a range similar to that in normal LX-2 cells; moreover, albumin secretion increased after treatment with 0.3 mM OA or Lipo-OAs.

Alcohol consumption is a critical risk factor for liver diseases; people who consume alcohol regularly have a high risk of several major chronic liver diseases. The main purpose of the current study was to evaluate whether airborne PM_2.5_ induces inflammatory responses in the mouse liver, and then whether long-term exposure to alcohol and PM_2.5_ synergistically increases liver inflammation or impairs liver function. In our in vivo experiments, we found the following: (1) Long-term exposure to PM_2.5_ was needed to induce any obvious liver inflammation or injury in our mice; (2) Exposure to PM_2.5_ and alcohol led to various liver inflammatory outcomes in our mouse model. Half of our mice died after the first alcohol induction and subsequent PM_2.5_ induction during the initial 3-week period of liver inflammation establishment. However, the mice who remained alive could withstand further continued inductions and treatments; (3) With exposure to both PM_2.5_ and alcohol, liver inflammation or injury was much more severe.

Our histopathological and blood biochemistry analyses indicated that after 4 and 6 weeks of treatment, Lipo-OAs had reparative effects on mouse livers with PM_2.5_ and alcohol-induced inflammation. Our Western blot analysis results also demonstrated that Lipo-OAs exhibited obvious anti-inflammatory effects by reducing αSMA, IL-6, and TNF-α levels (Figure 7). αSMA is a valuable marker for evaluating liver fibrosis progression and is an early indicator of fibrosis development. Here, we noted that exposure to PM_2.5_ alone or PM_2.5_ and alcohol increased the αSMA levels compared with no exposure—indicating that liver function became impaired gradually under the exposure. However, in histopathological observations, we did not observe a significant amount of liver fibrosis. Therefore, establishing a severe liver inflammation and fibrosis mouse model may require a longer PM_2.5_ or PM_2.5_ and alcohol exposure period. We also used Ishak’s modified HAI scores to examine the changes in livers with PM_2.5_- and PM_2.5_- and alcohol-induced inflammation for evaluating the reparative effects of OA and Lipo-OAs. Consistent with our other results, these scores demonstrated that Lipo-OAs have higher reparative efficacy than do OAs alone, or pure liposomes (Table 1, Table 2, Table 3 and Table 4).

Other reporters indicated that mice inhalation exposure to concentrated PM_2.5_-induced hepatic fibrosis by the increased expression of collagen in liver tissues. This hepatic fibrogenesis was via nicotinamide adenine dinucleotide phosphate (NADPH) oxidase mediation based on a mice model [6]. Ding et al. revealed that chronic exposure to ambient PM_2.5_ impaired oxidative homeostasis, caused inflammation, and induced oxidative injury and liver pathogenesis [33]. Combined PM_2.5_ exposure and high-fat diet apparently aggravated hepatic lipid metabolism disorders, inflammation, and progressed to non-alcoholic steatohepatitis. In our present study, we revealed that long-term, real-world ambient PM_2.5_ exposure and alcohol consumption led to more severe inflammatory liver responses in mice, which would equate more to the real situation stimulation in modern societies.

Several studies have reported oleanolic acid has antitumor, antimicrobial, antioxidant activities, anti-inflammatory potential, and hepatoprotective ability. However, there is little or literature evidence of pure OA itself as a candidate in clinical trials due to its toxicity. Recently, major progress has been made in the chemical modification of OA to achieve less toxicity and more bioavailable compounds. In this present study, we simply used liposome encapsulation techniques to achieve less toxicity and retained the reparative effect of Lipo-OAs to treat PM_2.5_-alcohol-induced inflammatory liver in mice. The results from LX-2 cells viability on OA and Lipo-OAs, HE staining, and blood biochemical analyses were all evident to reveal the lower toxicity and liver reparative effects of liposomal encapsulated OA [34].

Chronic liver inflammation would provoke hepatic fibrosis, and the persistent progression of liver inflammation would result in fibrosis, cirrhosis, and even hepatocellular-lar carcinoma [35]. Hepatic fibrosis can be prevented or reversed by eliminating the etiologic agents or disrupting the pathogenic mechanisms of liver injury. Currently, clinical trials in the treatment of hepatic fibrosis include inhibition of hepatocyte apoptosis, reduction of oxidative stress, inhibition of HSC activation, reduction of fibrotic scar evolution, and immune modulation [36]. For example, Silymarin is widely used to treat liver disorders, particularly in chronic liver diseases, cirrhosis, and hepatocellular carcinoma, because of its antioxidant, anti-inflammatory, and antifibrotic characteristics [37]. Song et al., also demonstrated that silymarin (200 mg/kg dosage) was able to reduce oxidative stress, as well as decreased ALT, Glutathione (GSH), lipid peroxidation, and to increase TNF-α levels [38]. In the present study, the obtained results from histopathological examination and blood biochemical analyses after treatment with Lipo-OAs showed similar therapeutic efficacy to silymarin, indicating that these Lipo-OAs have potential as a liver-protective and reparative agent against inflammation. However, it should be noted that mild liver inflammation was developed in the 3-w PM2.5 and alcohol induction period. Maybe a longer induction period was needed to cause more severe and obvious liver inflammation and injury and thereafter evaluate the therapeutic efficacy of OA and Lipo-OAs on these liver injuries.

## 5. Conclusions

Exposure to PM_2.5_ or PM_2.5-_-alcohol induced an inflammatory response in LX-2 cells as well as liver inflammation and injury in mice. Our liposomal encapsulation technique reduced OA cytotoxicity, and treatment with Lipo-OAs effectively alleviated the inflammatory response induced by PM_2.5_ or PM_2.5_–alcohol exposure. In summary, Lipo-OAs have potential as a liver-protective and reparative agent against inflammation.

## Figures and Tables

**Figure 1 pharmaceutics-14-01108-f001:**
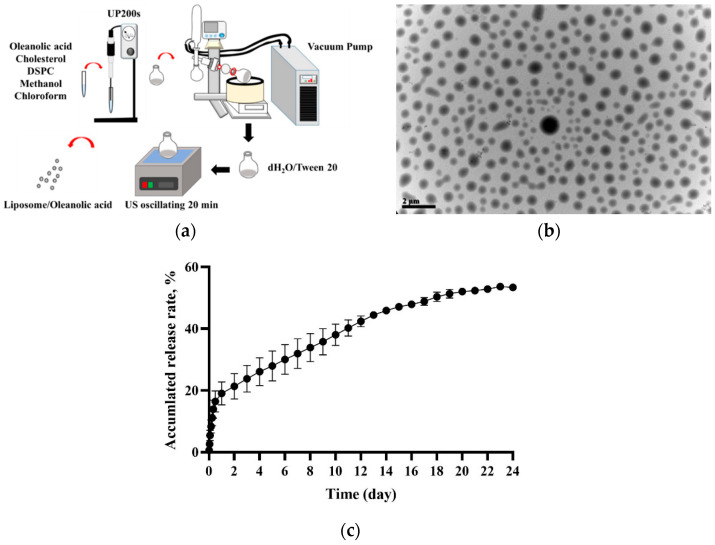
(**a**) Schematic of liposomal OA nanoparticle (Lipo-OAs) preparation; (**b**) Transmission election microscopy images of Lipo-OAs; (**c**) OA release profile of Lipo-OAs.

**Figure 2 pharmaceutics-14-01108-f002:**
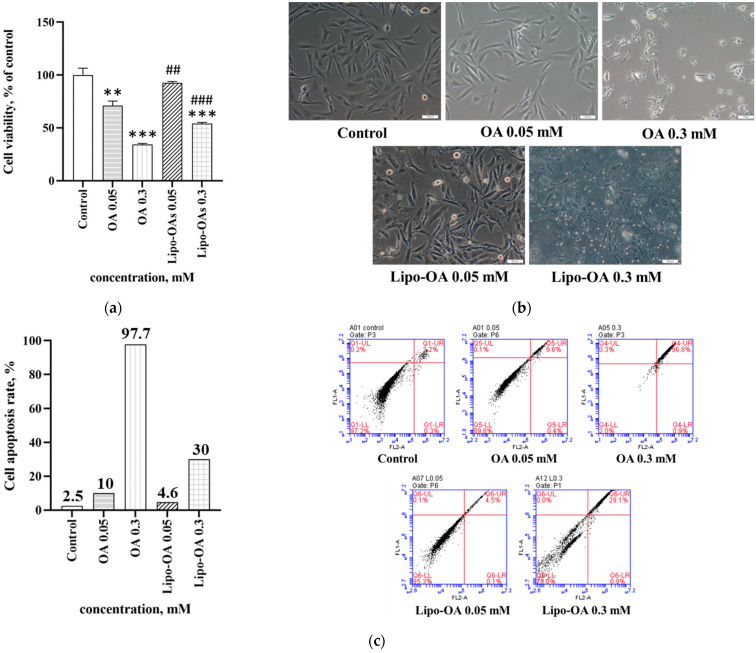
(**a**) MTT assay for OA and Lipo-OAs cytotoxicity; (**b**) Effects on LX-2 cell morphology monitored through light microscopy (magnification, 100×); (**c**) Flow cytometry assay for apoptosis in LX-2 cells treated with OA and Lipo-OAs at various concentrations. ** *p* < 0.01, *** *p* < 0.001 compared with control group. ## *p* < 0.01, ### *p* < 0.001 compared with OA. Solvent control: DMSO solution that used to dissolve OA.

**Figure 3 pharmaceutics-14-01108-f003:**
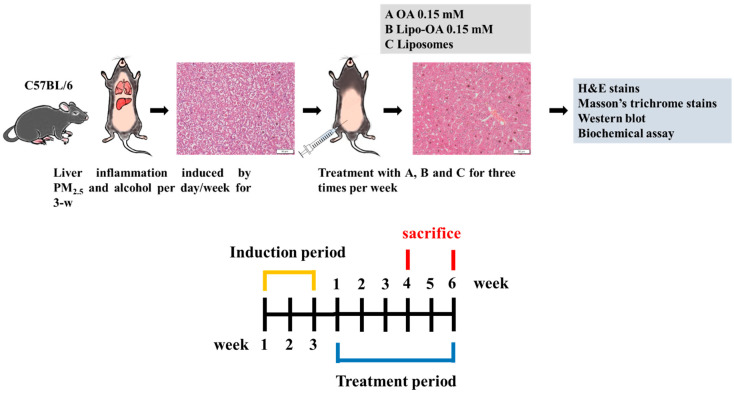
Schematic of PM_2.5_ and alcohol-induced liver inflammation model and treatment procedures.

**Figure 4 pharmaceutics-14-01108-f004:**
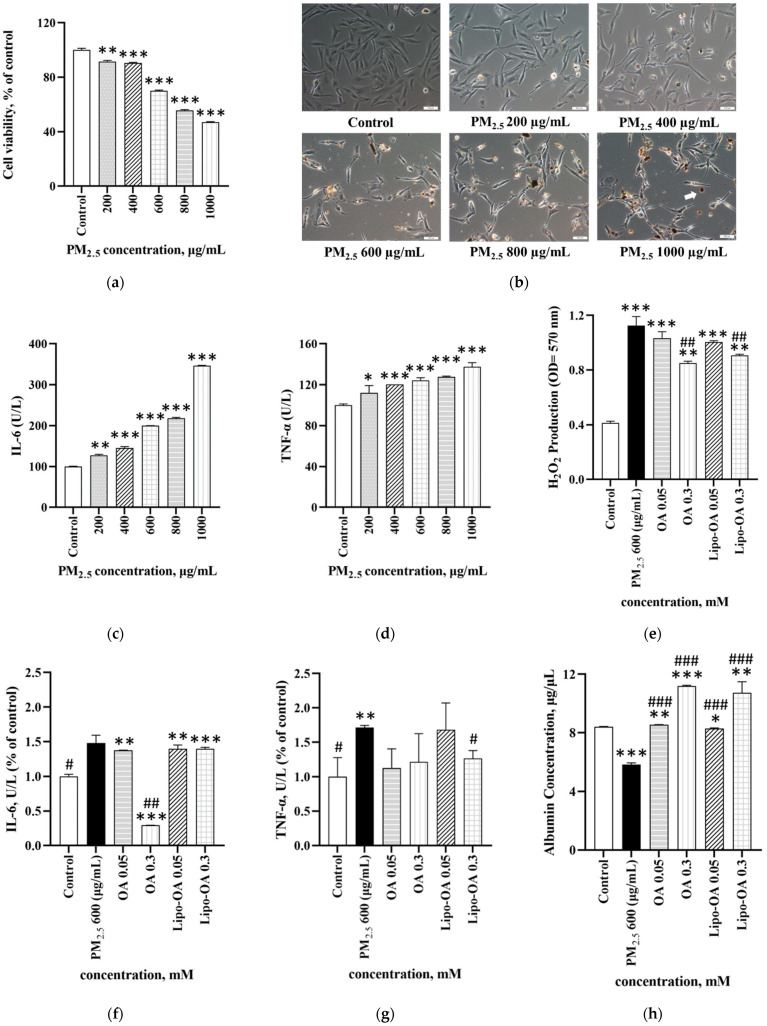
(**a**) MTT assay for PM_2.__5_ cytotoxicity; (**b**) Images of LX-2 cells treated with PM_2.5_ at various concentrations (bar: 200 μm); (**c**) IL-6; and (**d**) TNF-α response of LX-2 cells treated with PM_2.5_ at various concentrations; (**e**) Assessment of H_2_O_2_ scavenging activity; (**f**) IL-6; and (**g**) TNF-α response; and (**h**) Albumin secretion by LX-2 cells with PM_2.5_-induced inflammation after treatment with OA and Lipo-OAs at various concentrations. * *p* < 0.05, ** *p* < 0.01, *** *p* < 0.001 compared with control group; # *p* < 0.05, ## *p* < 0.01, ### *p* < 0.001 compared with PM_2.5_ 600 μg/mL group.

**Figure 5 pharmaceutics-14-01108-f005:**
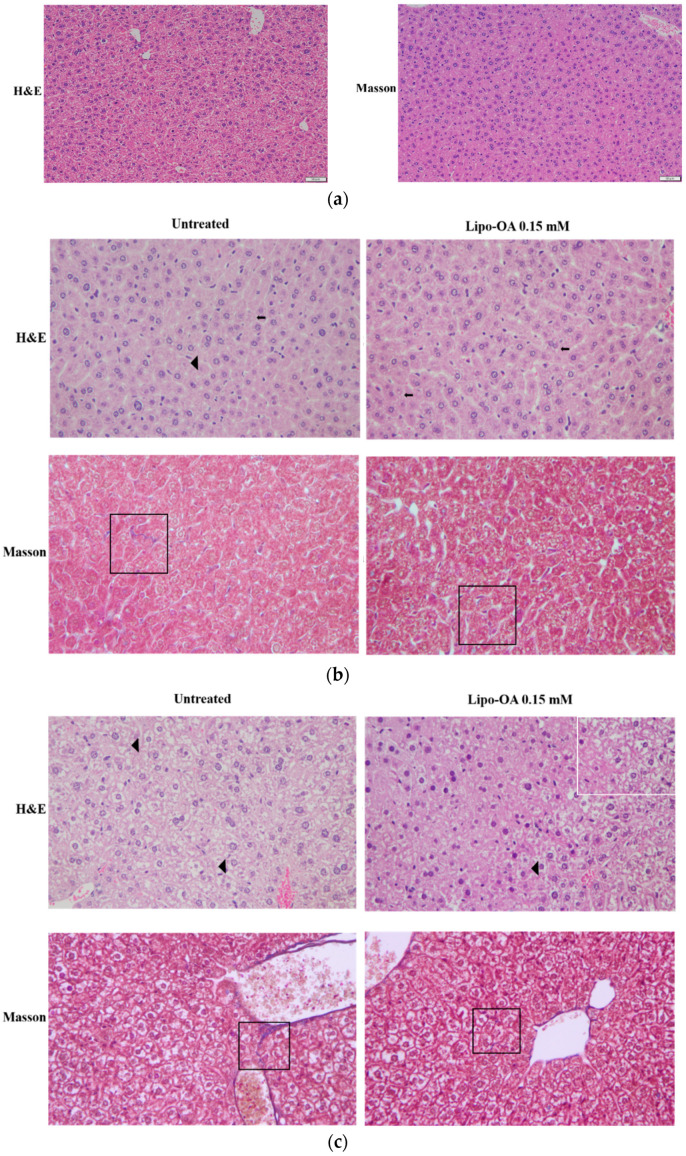
Hematoxylin and eosin (HE) staining and Masson’s trichrome staining of the sections of PM_2.5_-induced inflammation liver treated with Lipo-OAs (magnification, 200×): (**a**) normal; (**b**) after 4 treatment weeks; and (**c**) after 6 treatment weeks. The geometric figure was the representative area of: Patchy steatosis (white box); steatosis (black arrow); Mallory bodies and ballooning cells (black arrowhead); and pericellular fibrosis (black box).

**Figure 6 pharmaceutics-14-01108-f006:**
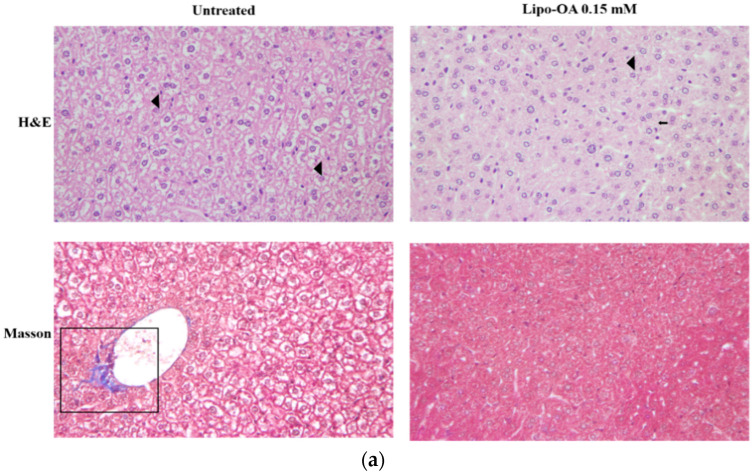
HE staining and Masson’s trichrome staining of the sections of PM_2.5_ and alcohol-induced inflammatory liver treated with Lipo-OAs (magnification, 200×): after (**a**) 4 and (**b**) 6 treatment weeks. The geometric figure was the representative area of: Patchy steatosis (white box); steatosis (black arrow); Mallory bodies and ballooning cells (black arrowhead); and pericellular fibrosis (black box).

**Figure 7 pharmaceutics-14-01108-f007:**
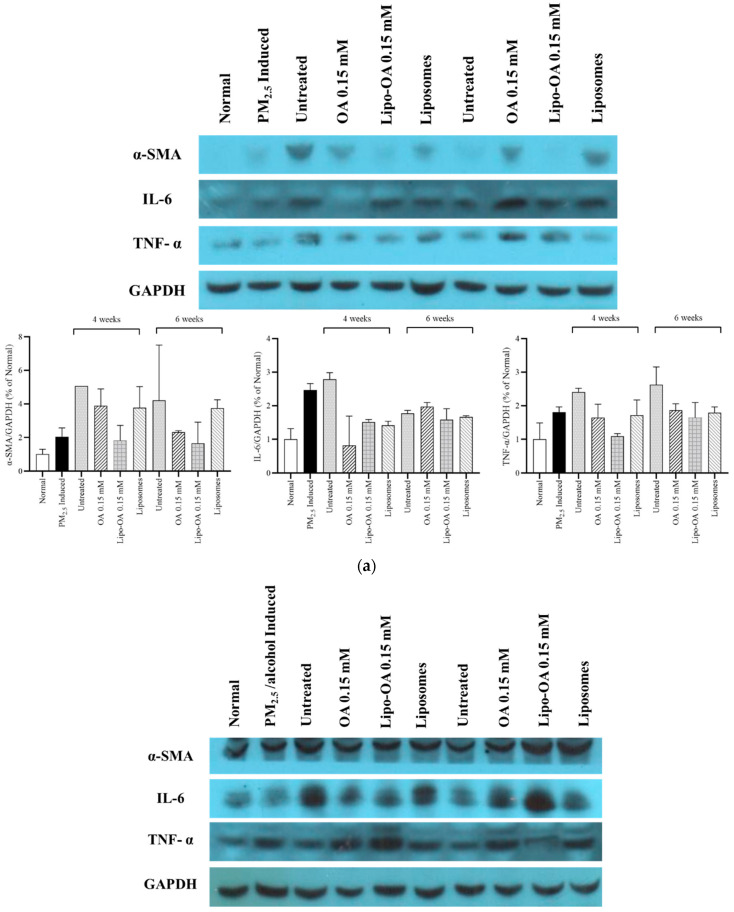
Western blot analysis for αSMA, IL-6, and TNF-α in livers with (**a**) PM_2.5_-induced inflammation; and (**b**) PM_2.5_ and alcohol-induced inflammation after treatment with or without 0.15 mM OA, 0.15 mM Lipo-OAs, or pure liposomes. The semiquantitative levels of αSMA, IL-6, and TNF-α were determined using ImageJ (n = 2).

**Figure 8 pharmaceutics-14-01108-f008:**
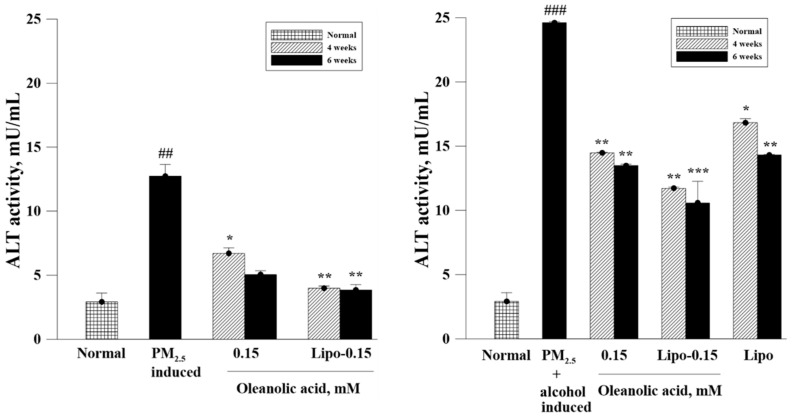
Effects of OA and Lipo-OAs on ALT, AST, and GGT levels in the mice with PM_2.5_ or PM_2.5_- and alcohol-induced inflammation in the liver. * *p* < 0.05, ** *p* < 0.01, *** *p* < 0.001 compared with induced group; ## *p* < 0.01, ### *p* < 0.001 compared with normal group.

**Table 1 pharmaceutics-14-01108-t001:** Histological findings and scores after 4 treatment weeks in PM_2.5_-induced inflammation liver.

Group	Steatosis	Mallory Body and Ballooning Cells	Ishak Modified HAI Score	Pericellular Fibrosis
untreated	+	++	1	++
OA	+	++	0	+~++
Liposome	++	++	2	++
Lipo-OAs	+	+	0	+

Percentage of steatosis (steatosis area/total liver area in the slide): +++ (>66%); ++ (33~66%); + (0~33%); Degree of pericellular fibrosis: +++ (diffuse); ++ (patchy); + (focal); −/+ (very focal); Quantity of Mallory body and Ballooning cells: +++ (many); ++ (some); + (few); − (absent).

**Table 2 pharmaceutics-14-01108-t002:** Histological findings and scores after 6 treatment weeks in PM_2.5_-induced inflammation liver.

Group	Steatosis	Mallory Body and Ballooning Cells	Necrosis	Ishak Modified HAI Score	Pericellular Fibrosis
untreated	+++	+++	−	1	−/+
OA	+++	+++	+	3	+
Liposome	+++	+++	+	2	+
Lipo-OAs	++	+~++	−	0	−/+

Percentage of steatosis (steatosis area/total liver area in the slide): +++ (>66%); ++ (33~66%); + (0~33%); Degree of pericellular fibrosis: +++ (diffuse); ++ (patchy); +(focal); −/+ (very focal); Quantity of Mallory body and Ballooning cells: +++ (many); ++ (some); + (few); − (absent); Range of necrosis: +++ (patchy); ++ (mild); + (focal).

**Table 3 pharmaceutics-14-01108-t003:** Histological findings and scores after 4 treatment weeks in PM_2.5_ and alcohol-induced inflammation liver.

Group	Steatosis	Mallory Body and Ballooning Cells	Ishak Modified HAI Score	Pericellular Fibrosis
Untreated	+++	+++	2	−/+
OA	+++	+++	1	−/+
Liposome	+++	+++	2	−/+
Lipo-OAs	+	+	0	−/+

Percentage of steatosis (steatosis area/total liver area in the slide): +++ (>66%); ++ (33~66%), + (0~33%); Degree of pericellular fibrosis: +++ (diffuse); ++ (patchy), +(focal); −/+ (very focal); Quantity of Mallory body and Ballooning cells: +++ (many); ++ (some); + (few); − (absent).

**Table 4 pharmaceutics-14-01108-t004:** Histological findings and scores after 6 treatment weeks in PM_2.5_- and alcohol-induced inflammation liver.

Garoup	Steatosis	Mallory Body and Ballooning Cells	Ishak Modified HAI score	Pericellular Fibrosis
Untreated	+++	+++	2	−/+
OA	+++	+++	1	−/+~+
Liposome	+++	+++	2	−/+
Lipo-OAs	+++	++	0	−/+

Percentage of steatosis (steatosis area/total liver area in the slide): +++ (>66%); ++ (33~66%); + (0~33%); Degree of pericellular fibrosis: +++ (diffuse); ++ (patchy); +(focal); −/+ (very focal); Quantity of Mallory body and Ballooning cells: +++ (many); ++ (some); + (few); − (absent).

## Data Availability

The data presented in this study are available in the paper.

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
