# Peer review of "Reparative Efficacy of Liposome-Encapsulated Oleanolic Acid against Liver Inflammation Induced by Fine Ambient Particulate Matter and Alcohol in Mice"

_pharmaceutics, 2022, doi:10.3390/pharmaceutics14051108_

Round 1
Reviewer 1 Report
The paper submitted by Wei et al. deals with the preparation of oleanolic acid-loaded liposomes and the study of their efficiency against the liver inflammation of mice produced by a combination of two factors, such as: fine ambient particles and alcohol.
The paper is clear, well written and the conclusions are supported by the results. However, some corrections are needed in order to increase the overall quality of the paper:
- in the introduction section some references must be added for the paragraph dealing with the description of liposomes. Some suggestions are: https://doi.org/10.3390/polym11091515; https://doi.org/10.3390/pharmaceutics13060866.
- Also a reference must be added in paragraph 2.5 at the end of the sentence: "An albumin secretion assay...as described elsewhere."
- Could the authors explain why they haven't reach the equilibrium for the release of OA?
- the discussion section is quite poor. Generally, in this section the authors must discuss their results in comparation with existing literature data. In the present form, the discussion section is only an extended conclusion section.
Author Response
Please see the attached file, thanks!

Reviewer 2 Report
Referee’s comments regarding the Manuscript ID: pharmaceutics-1701380, entitled
Reparative Efficacy of Liposome-Encapsulated Oleanolic Acid Against Liver Inflammation Induced by Fine Ambient Particulate Matter and Alcohol in Mice
by Ching-Ting Wei, Yu-Wen Wang, Yu-Chiuan Wu, Li-Wei Lin, Chia-Chi Chen, Chun-Yin Chen and Shyh-Ming Kuo.
The manuscript describes the in vitro and in vivo investigations of biological impact and reparative efficacy of liposome-encapsulated oleanolic acid (OA) against the liver inflammation. The topic of the manuscript is a very interesting especially because it relates to formulation of very demanding hidrophobic OA with a variety of potential biological functions.
The first remark of the reviewer on this manuscript concerns the fact that the authors of the manuscript neglected to cite relevant papers describing previously prepared liposomal formulation of OA and the results of its biological testing (e.g. D. Gao et al, 2012; S. Tang et al., 2013; M. Sarfraz et al., 2017.)
The second objection refers mainly to the figures in the manuscript. From the reviewer’s viewpoint, some figures need to be rearranged. Due to the overfilling of the images, the effects cannot be clearly seen. In particular, Figures 2 and 3 where results of MTT assay are presented, should be redesigned, most of the results presented in the Figure 2 should be moved to supplementary materials and only the effect of min and max doses should be shown. Figures 5 and 6 where patohistological effects are presented should also be revised and the authors should select key images while the others should relocated to the supplement.
Conclusion: The manuscript brings new results in the field of potential application of the liposome formulation of OA and it could appeal to a broad readership of the Pharmaceutics journal.
Therefore, I would recommend the publication of this manuscript after the authors make the necessary revisions in the manuscript.
Author Response
Please see the attached file, thanks!

Round 2
Reviewer 1 Report
The paper can be published as it is.
Reviewer 2 Report
The authors made corrections in the manuscript and answer the reviewer’s comments. These corrections improved the manuscript.
I recommend the publication of the manuscript in the Pharmaceutics in this form.